



# Aeration and mineral composition of soil determine microbial CUE

Jolanta Niedźwiecka[1,2], Roey Angel[1,2,4], Petr Čapek[2], Ana Catalina Lara[1,3], Stanislav Jabinski[1,2], Travis B. Meador[1,2], Hana Šantrůčková[2,4]

5    1. Institute of Soil Biology and Biogeochemistry, Biology Centre CAS, České Budějovice, Czechia

2. Faculty of Science, University of South Bohemia in České Budějovice, Czechia

3. Present address: Department of Biochemistry and Microbiology, University of Chemistry and Technology, Praha, Czechia.

4. Correspondence: roey.angel@bc.cas.cz or hasan@prf.jcu.cz

10    **Abstract**

Microbial carbon use efficiency (CUE) in soils is used to estimate the balance of $CO_2$ respired by heterotrophs versus the accumulation of organic carbon (C). While most CUE studies assume that aerobic respiration is the predominant degradation process of organic C, anoxic microniches are common inside soil aggregates. Microorganisms in these microniches carry out fermentation and anaerobic respiration using alternative electron acceptors, e.g. $NO_3^-$,

Fe, $SO_4^{2-}$. Extracellular metabolites are also not traditionally accounted for but may represent a significant C flux. Moreover, climate change may modulate soil microbial activity by altering soil aeration status on a local level due to warming and elevated frequency of extreme precipitation events. Therefore, CUE should be measured under more realistic assumptions regarding soil aeration. This study focused on the effect of oxygen and Fe on C mineralisation in forest soils and quantified C distribution between biomass and different extracellular metabolites. Forest soils

were collected from two Bohemian Forest (Czechia) sites with low and high Fe content and incubated under oxic and anoxic conditions. A solution of 13C-labelled glucose was used to track stable isotope incorporation into the biomass, respired $CO_2$, and extracellular metabolites. We estimated CUE based on microbial respiration, glucose consumption, biomass growth, and extracellular metabolites. RNA-SIP was used to identify the active bacteria under each treatment. As expected, the oxic incubation showed a rapid utilisation and immediate production of biomass and $CO_2$. Under anoxic conditions, 90% of the added glucose was still present after 72 h, and anoxic soils showed

significantly lower microbial activity. The low-Fe soil samples were more active under oxic conditions, while the high-Fe samples were more active under anoxia.

Our findings confirm that anoxia in soils enhances short-term C preservation. Accordingly, excluding exudates in mass flux calculations would underestimate apparent CUE values.

**1 Introduction**

In soils, the efficiency of microbial transformation of plant litter to soil organic carbon (SOC) is one of the main controls of carbon (C) storage. Carbon use efficiency (CUE) is a significant factor that affects the potential for C storage in soils and depends on partitioning into biogeochemical compartments (Tao et al., 2023). CUE is especially helpful



for describing changes in the C soil storage due to shifts in climate patterns and, consequently, in environmental con-

ditions. CUE is also useful for predicting C transformations in terrestrial ecosystems, offering a glimpse into their

functioning and, subsequently, the effect of ecosystem management strategies.

Traditionally, CUE is calculated as biomass production over organic C uptake (e.g. Manzoni et al., 2012; Sinsabaugh et al., 2013). In this sense, gross CUE equals microbial growth efficiency (del Giorgio and Cole, 1998). High CUE means that C is stored in the soil as microbial biomass and, over time, can be stabilised as SOC. Low CUE means or-

ganic C is used in catabolism and released into the atmosphere as $CO_2$ or $CH_4$. By most working definitions, CUE assumes biomass formation and $CO_2$ production as fates for C substrates. Hence, this definition excludes extracellular metabolites (cell exudates) such as organic acids, alcohols, enzymes, extracellular polymeric substances, etc., that can also increase soil potential for C sequestration (Manzoni et al. 2012). Even under fully oxic conditions, a significant release of microbial exudates (sugars and amino acids) has been documented in addition to long-lasting necro-

mass components such as muramic acid (Warren, 2022). In humid or dry climates, soils contain pockets where anoxic metabolism occurs (Ebrahimi and Or, 2015; Endress et al., 2024). Under anoxia, microbial exudates can be retained in the soil and, like necromass, become a component of stable soil organic matter (SOM). In oxic soils, enzyme production rate is estimated to account for 1 to 5% of the microbial biomass (Allison et al., 2010; Allison, 2014). However, in anoxic soils and anoxic microniches of well-aerated soils, the production of extracellular organic

compounds in "hot spots of activity" or aggregates (Borer et al., 2018) can be comparable in magnitude with the total microbial biomass (Picek et al., 2000; Šantrůčková et al., 2004). Thus, microbial biomass and exudate production should be included in CUE calculations, especially when CUE is applied to estimate the conversion of plant litter to SOM and, ultimately, C storage potential in soil. Under some circumstances, neglecting microbial exudation could lead to underestimating the role of microbial organic matter transformation in increasing the C storage in soil over a

certain period. In the following text, we will discuss the effect of aeration status on microbial biomass and exudate production and compare different approaches to estimate CUE. To avoid ambiguity, we employ the terminology proposed by (Manzoni et al., 2018) and use the term apparent carbon use efficiency ($CUE_A$) when only biomass production is included in the calculation and CUE if microbial exudates (e.g. enzymes, fermentation products, polysaccharides) are also included. In addition, we define the carbon storage efficiency (CSE), which includes both biotic and

abiotic C accumulation, and carbon stabilisation efficiency (TCSE), which measures the abiotic accumulation of C in soil (see formal definitions below and in Table S1). Both definitions above are influenced by substrate chemical composition, stoichiometry (Blagodatskaya et al., 2014; Manzoni et al., 2018), microbial functional diversity, and many climatic and edaphic drivers. Of those, the availability of electron acceptors and associated redox conditions play the most important role in controlling the gap between $CUE_A$ and CUE (and naturally also CSE). Since SOC is

used for both energy metabolism and biosynthesis, biomass production (growth yield) closely matches energy gain from dissimilatory reactions, either in aerobic and anaerobic respiration or fermentation (Roden and Jin, 2011).

Under aerobic and anaerobic respiration, $CO_2$ is the main end-product of dissimilation, and microbial biomass is the main product of biosynthesis. Although the biomass yield decreases with decreasing energy gain in anaerobic respiration depending on the availability of electron acceptors ($NO_3 > Mn^{4+} > Fe^{3+} > SO_4^{2-}$), when dissimilation pathways

prevail in soil, the difference between $CUE_A$ and CUE should not be large. However, the close relationship between biomass synthesis and energy gain fails under certain environmental conditions. For example, when degrading lignocellulose, many of the enzymes involved in the processes are exoenzymes and can remain in the soil matrix (Gotsmy et al., 2021). Under desiccation stress, prokaryotes secrete extracellular polymeric substances (EPS; mostly polysaccharides and amino acids to limit water loss) (Kakumanu and Williams, 2019). Under unbalanced growth with nutri-



ent deficiency, overflow metabolism can occur, and cells release excess C from dissimilatory processes (Dijkstra et al., 2022; Basan et al., 2015). However, such fluxes are also typically not accounted for when measuring biomass production. Lastly, when external electron acceptors are depleted in anoxic conditions, and fermentation pathways prevail as energy metabolism, growth yields are much lower, and a significant portion of the carbon is secreted back into the soil (Liu et al., 2007). Under anoxic conditions, when fermentation will diverge significantly from CUE and

CSE.

One of the most abundant electron acceptors that play a key role in anoxic environments is Fe(III). Iron-reducing microorganisms (FRM) are ubiquitous in soil and can be found in various taxa across the phylogenetic tree in both Bacteria and Archaea (Weber et al., 2006). In soils, FRMs gain access to iron-oxide crystals either via direct contact, nanowires (Reguera et al., 2005) or through organic electron shuttles such as quinones (formerly termed collectively

with other substances as "humic substances"; Bauer and Kappler, 2009; Newman and Kolter, 2000) and use them to oxidise organic compounds in anaerobic respiration. However, in addition to serving as a terminal electron acceptor in anaerobic respiration, Fe(III) can serve as a sink for excess reducing equivalents during fermentation, thereby reducing $H_2$ or ethanol production (Lovley et al., 2004). Fermenters are particularly efficient in breaking down more labile organic C sources, such as sugars, amino acids, and fatty acids, compared to FRM (Lovley, 2014). This again

shows that extracellular products of anaerobic metabolism may significantly differ from aerobic metabolism and that careful determination of the C budget under both conditions is needed to calculate realistic CUE values.

As mentioned above, CUE reflects the substrate quality and stoichiometry, microbial structural diversity, environmental factors, and a methodological approach (Geyer et al. 2019). For example, the mechanistic approach of measuring at the same time interval, which is usually used, may introduce some bias because $CUE_A$ decreases over time

due to the gradual transformation of the consumed substrate and its loss in the form of $CO_2$ or extracellular compounds. This decrease depends on the physiological state or growth of the microbial population and, thus, on the environmental conditions during the experiment. This should be considered when conducting experiments in different environmental conditions and if we want to know not only $CUE_A$ (growth yield) but also the CUE, CSE or TCSE. Measurements should be made at different time intervals and variants with microbial populations in similar growth

stages compared.

To assess the effect of aeration status and the availability of iron as an alternative electron acceptor on $CUE_A$, CUE, CSE, and TCSE, we employed a suite of chemical and stable isotope labelling methods to measure the soil properties and CUEs. Simultaneously, we identified the active members of the community using RNA-stable isotope probing (RNA-SIP). Combining CUE measurements with RNA-SIP links the ecosystem function of microbial populations

with the taxonomic identity of the active bacteria.

We studied two acidic forest soils with similar chemical and biochemical properties but differing in Fe content. We hypothesised that: 1. Oxic $CUE_A$ will be much higher than anoxic while the difference between oxic and anoxic CUE is low. 2. Under oxic conditions, C is used mostly for respiration and biomass production, while under anoxic conditions, the production of exudates, such as organic acids, is important, resulting in a divergence between $CUE_A$ and

CUE. 3. Fe content affects the divergence between $CUE_A$ and CUE and C storage and stabilisation efficiencies. Low iron soils have a faster onset of fermentation. 4. Different microbial communities will be isotopically enriched under oxic and anoxic conditions. Iron content will also affect the community profile of the active microbes in anoxic soils. To test these hypotheses, we incubated two soils with different Fe content under oxic and anoxic conditions and labelled them with $^{13}$C-glucose. Labelled glucose partitioning into gaseous products ($CO_2$/$CH_4$), microbial biomass, ex-



udates, and nonextractable organics were determined at different time intervals of incubation. RNA-SIP was employed to identify the microbes involved in carbon assimilation.

## 2 Methods

### 2.1 Site description and soil characteristics

Spruce forest soils were collected on two sites in the Bohemian Forest, southern Czechia. The soils at both sites are
acidic ($pH_{H2O}$ varies between 3 and 4) and chemically similar, apart from iron oxide content (Kaňa et al., 2019) (Table S2). Soil from Plešné Lake (PL; 48° 46' 35 "N, 13° 51' 53 "E) catchment area contains ca. 35 mM $Fe_{tot}$ kg$^{-1}$ and soil from Čertovo Lake (CT; 49° 9' 54 "N, 13° 11' 47" E) catchment area contains ca. 240 mM $Fe_{tot}$ kg$^{-1}$. Both Fe values represent oxalate-extractable Fe(III). Soil was sampled from the top 5 cm of organomineral layers (A horizons) by pooling soil from four randomly chosen locations. Fresh soils were sieved (2 mm) and homogenised prior
to storage at 4 °C for several days before the experiments.

### 2.2 Experimental setup

Two incubation setups of the same design were run in parallel: *(i)* A high-volume, low-level, $^{13}$C-labelling incubation to measure C balance and C partitioning into different metabolites and *(ii)* a low-volume, high-level $^{13}$C-labelling incubation for identifying active bacteria and archaea using RNA-SIP. The moisture of both soils was adjus-
ted to 50% of the water-holding capacity to ensure the same hydration conditions. Fresh soil samples, weighing 32 or 3.9 g (see below) were transferred into 250 ml or 30 ml glass bottles, sealed with preboiled butyl rubber stoppers, and preincubated for 6 days at 20 °C in either oxic (lab air, always 16 bottles) or anoxic ($N_2$, always 16 bottles) conditions. Anoxic conditions were established by flushing the headspace with $N_2$ gas and storing the bottles upside down, with caps immersed in water, to limit oxygen infiltration through the septa.

### 2.2.1 Soil incubations for C balance and chemical analyses

After 6 days of preincubations, all bottles were opened to add 8 ml of a $^{13}$C-labeled-glucose solution to a final glucose concentration of approximately 100 µg C g$^{-1}$ of dry weight and AT% $^{13}$C of 3%. The glucose solution was prepared by mixing 99 AT% glucose (uniformly labelled, Sigma-Aldrich) with natural $^{13}$C abundance glucose (Sigma-Aldrich). The spike amendment increased the moisture in both soils to 64-67% in PL and 57-58% in CT. Anoxic
bottles were handled inside an anoxic vinyl chamber (COY) filled with 98% $N_2$ and 2% $H_2$, and the isotope solution was degassed using a gassing manifold before adding it to the experimental bottles. Soils were incubated at 20 °C for 72 h under oxic conditions for measurable microbial growth and glucose transformation. Under anoxic conditions, the soil was incubated for 216 h because of slow anaerobic metabolism and long lag phase. Oxygen was monitored throughout incubation. In the oxic bottles, $O_2$ only slightly decreased during the incubation, and in the anoxic bottles,
no $O_2$ was detected. Headspace gas ($CO_2$ and $CH_4$) and C isotopes were measured at 0, 24, 48, and 72 (oxic) or 216 h (anoxic) of incubation and then, soils were sampled destructively (always 4 replicates) for biochemical and chemical analyses. Soils for total C analyses were freeze-dried immediately after sampling. Unamended controls were set up and incubated like experimental samples, except that water was added to adjust moisture instead of glucose solution.



Headspace gas in the controls without glucose addition was measured in the same period as glucose-amended samples. However, destructive analyses were performed only at the beginning and end of incubation (0 and 72 or 216 h).

### 2.2.2 Soil incubations for RNA-SIP

On a smaller scale (30 ml bottles), separate soil incubations were performed with the same concentration of 99% uniformly $^{13}C$-labelled glucose for RNA-SIP analysis. A high glucose label was required for effective $^{13}C$ incorporation into nucleic acids, a prerequisite for SIP. The incubations were set up in the same way as the high-volume incubations described above. Most importantly, the soil-to-headspace volume ratio was identical in both incubation setups and $CO_2$ and $CH_4$ production was monitored throughout the experiment to ensure comparable experimental conditions. Oxic and anoxic samples were collected periodically in the same time intervals described above and immediately frozen on dry ice until further processing.

### 2.3 Chemical and biochemical analyses

#### 2.3.1 Headspace gases

Concentrations of $CO_2$, $CH_4$, and $O_2$ were measured using an HP 6850 gas chromatograph (Agilent) equipped with a 0.53 mm × 15 m HP-Plot Q column, a 0.53 mm × 15 m HP-Plot Molecular Sieve 5A column and a thermal conductivity detector, with He as the carrier gas. CH4 was measured using an HP 6890 gas chromatograph (Agilent) equipped with a 0.53 mm x 30 m GS-Alumina column and a flame ionisation detector, with $N_2$ as the carrier gas. Peaks were integrated using Agilent Chemstation A.08.03 software (Agilent). The $^{13}C$-CO2 fraction of the headspace was measured by injecting 250 µl of headspace sample to the gas preparation and introduction system Delta/MAT 252 Gasbench II (Thermo Finnigan) connected to Delta$^{plus}$XL IRMS (Thermo Finnigan), with He as a carrier gas. The precision for measurements was <0.2‰ for peaks >3000 mV.

#### 2.3.2 Total C, N and $^{13}C$ in various pools

Unless mentioned otherwise, all concentrations of soil chemical parameters are expressed per g of dried soil. Bulk soil N ($N_{tot}$), C ($C_{tot}$) and its isotopic signal ($δ^{13}C_{tot}$) were determined in freeze-dried grounded soil samples. Water extractable N (TN), nitrates ($NO_3^-$), C and $δ^{13}C$ (WEC) were determined after a 1 h extraction with distilled water (1:10, w/v) on a shaker, and filtration (glass-fibre filter, 0.45 µm). Carbon and $δ^{13}C$ in microbial biomass carbon ($C_{MB}$) were estimated by the chloroform-fumigation extraction method (Vance et al., 1987) modified for $^{13}C$ analysis (Bruulsema and Duxbury, 1996). Briefly, non-fumigated or fumigated ($CHCl_3$, 24 h) soil was extracted with 0.05 M $K_2SO_4$ (1:4, w/v), and the extract was centrifuged and filtered through a glass filter (0.45 µm). The extracts in potassium sulphate (fumigated and non-fumigated) and in water (WEC) were freeze-dried for a final dry C weight of approximately 30 µg, resuspended in distilled-deionised water (ddH₂O), quantitatively transferred 600 µl into tin cups (10 x 10 mm, Sercon) and dried overnight in the oven at (40 °C). $C_{MB}$ was calculated as the difference between C in the fumigated and non-fumigated soil extracts, assuming the relative extractability of microbial cells killed by fumigation ($K_{EC}$) to be 0.38 (Vance et al., 1987). The $δ^{13}C$ of $C_{MB}$ was calculated by a two-component mixing model (Šantrůčková et al., 2000). Total dissolved N and organic C dissolved in $K_2SO_4$ or water were analysed on a



TOC/TN analyser (LiquicTOC II, Elementar, Germany). Freeze-dried bulk soil and extracts were analysed via flash
175   combustion at 1020 °C on a SmartEA Isolink with a continuous flow interface to a Conflo IV device and MAT253
Plus IRMS (Thermo-Fisher Scientific). Values of $\delta^{13}C$ were determined against an international reference (IAEA-
600 and -603) and house standards and are expressed in $^{13}C$ atom per cent. The precision of the $\delta^{13}C$ measurements
for natural abundance samples was 0.06‰. All analyses were performed in four replicates unless stated otherwise.
Bulk N ($N_{tot}$) was measured in the same samples as for C and $^{13}C$ analyses using an EA Isolink. Water extractable
$NO_3^-$ was measured spectrophotometrically with a flow injection analyser (FIAstar 5012, Foss Tecator, Sweden).
Glucose concentration remaining in the soil was monitored at each sampling point in four replicates as described pre-
viously (Šantrůčková et al., 2004; Picek et al., 2000). Glucose was extracted from 10 g of soil sample using 15 ml of
0.1% (w/v) benzoic acid. The soil solution was shaken on an end-over-end shaker at 150 rev min$^{-1}$ for 15 min and
centrifuged at 7000 ×g for another 15 min. Proteins were precipitated with a final concentration of 10% (w/v) tri-
chloroacetic acid (Koontz, 2014). Glucose concentration was measured enzymatically using the BIOLATEST assay
GLU 500 (Erba Lachema) and expressed per C1-molar basis ($C_{gluc}$).

### 2.3.4 Bioavailable Fe(II) and Fe(III)

Bioavailable Fe(II) ($Fe_{avail}$) concentrations were measured using the ferrozine assay (Lovley and Phillips, 1987).
Briefly, 1 g of fresh soil was submerged in 5 ml of 0.5 N HCl for 10 min. Then, the acid extract was added to a fer-
rozine solution, and Fe(II)-ferrozine complex was measured spectrophotometrically at 562 nm. Fe(III) in the acid ex-
tract was reduced to Fe(II) using hydroxylamine-HCl, and total Fe(II) was measured as described above. The differ-
ence between total Fe and Fe(II) was equal to bioavailable (microbially reducible) Fe(III) in the acid extract. Repor-
ted Fe(II) values are likely underestimated, as soil aliquots came into contact with oxygen after sampling but before
acidification and Ferrozine assay. For this reason, Fe(II) values are discussed comparatively rather than as absolute
values.

### 2.3.5 Organic acids in pore water

Sample porewater was analysed to measure low molecular weight organic acids (OA) production via fermentation.
The porewater was separated from moist soil by centrifugation. For CT soil, 8 g of moist soil was centrifuged at
8000 × g for 10 min. For PL soil, 14 g were centrifuged at 10 400 × g for 10 min. Centrifuged liquids were analysed
by ion chromatography (Integrion, Thermo Fisher) with a conductivity detector. Organic acids were separated using
IonPac AG11-HC-4 and IonPac AS11-HC-4 columns (Thermo Fisher). The eluent was analytical EGC KOH with a
multistep gradient ranging from 1 mM to 85 mM and run at 0.38 ml min-1. Samples were placed in a cooled (5 °C)
AS-AP autosampler (Thermo Fisher). Samples of 15 µl with 20 µl cut volumes were injected using low draw speed.
Samples were run across certified standards (Sigma-Aldrich).

### 2.4 Identification of labelled bacteria using RNA stable-isotope probing

### 2.4.1 Nucleic acid extraction, RNA purification

RNA was extracted from each sample from the small-scale incubation, as described before (Angel et al., 2011).
Briefly, total nucleic acids (TNA) were extracted by disrupting 0.2 g of soil from each sample in a lysing matrix E



tube (MP Biomedicals) in a chilled environment (dry ice), in the presence of phosphate buffer, 10% SDS solution,
phenol and 0.1 M AlNH$_4$(SO$_4$)2×12H2O (Alfa Aesar) using a FastPrep–24™ 5G sample homogeniser and Cool-
Prep™ adapter (MP Biomedicals). The process was repeated 3 times for each sample, using the same lysing matrix
tubes, collecting the supernatant after each time, and using fresh buffers and phenol. TNA was then purified using
phenol/chloroform/isoamyl alcohol and chloroform/isoamyl alcohol purification (Carl Roth). TNA was precipitated
using 30% polyethylene glycol and 2 µl of glycogen (Life Technologies), washed once with ice-cold, 75% EtOH,
and resuspended in low TE buffer in Non-Stick RNase-free Microfuge Tubes (Life Technologies). Lastly, TNA was
further purified using the OneStep PCR inhibitor removal kit (Zymo Research). For DNA removal, 90 µl of TNA
was digested with TURBO DNase (Life Technologies) and later purified using GeneJET RNA cleanup and concen-
tration micro kit (Thermo Fisher Scientific). Complete DNA removal was verified by failure to obtain a PCR ampli-
fication product using the purified RNA template; the PCR conditions are described below. The purified RNA was
quantified using Quant-iT RiboGreen RNA Assay Kit (Thermo Fisher Scientific). The full protocol is available on-
line (Angel et al., 2021).

### 2.4.2 RNA-stable isotope probing

RNA (ca. 250 ng) was subjected to isopycnic gradient centrifugation in a solution of caesium trifluoroacetate
(CsTFA, GE Healthcare), HiDi formamide (Thermo Fisher Scientific) and buffer (0.1 M Tris-HCl at pH 8.0, 0.1 M
KCl and 1 mM EDTA). Gradients were prepared in Ultracrimp 6 ml tubes (Thermo Scientific) and centrifuged at
130 000 × g using a TV-1665 Sorval Rotor (Thermo Scientific) for ≥65 hours. Fractionation was done by piercing
the tube close to the top and bottom and injecting RNAse-free water (Carl Roth) to the top using an automatic syr-
inge pump (NE-300; New Era Pump System Inc.) Fractions were collected every 20 s (300 µl each). The density of
each fraction was measured using a refractometer (AR200 Automatic Digital Refractometer; Reichert). Fifteen (15)
fractions out of 20, with densities ranging between 1.766-1.842 g ml$^{-1}$, were used for downstream analysis. The RNA
from the fractions was precipitated in the presence of 2 µl GlycoBlue (Fisher Scientific), Na-acetate (3 M) and EtOH
(absolute). RNA pellets were dissolved in 10 µl of RNA-Storage solution (Fisher Scientific). Complementary DNA
was synthesised using Super Script IV reverse transcriptase (Thermo Fisher Scientific) and 0.5 µg µl$^{-1}$ of random
hexamer primers, as described by the manufacturer. The full protocol is available online (Angel and Petrova, 2021).

### 2.4.3 Amplicon sequencing

Amplicon sequencing was done using a two-step barcoding approach (Naqib et al., 2018). cDNA from each fraction
was amplified using the universal bacterial and archaeal primers 515F-mod-CS1 (aca ctg acg aca tgg ttc tac aGT
GYC AGC MGC CGC GGT AA), 806-mod-CS2 (tac ggt agc aga gac ttg gtc tGG ACT ACN VGG GTW TCT AAT;
Walters et al., 2016) in a T100 Thermal Cycler (Biorad) with a number of amplification cycles ranging from 26 to
30, depending on the amount of template. The full protocol is available online (Angel and Petrova, 2021). In addi-
tion, negative control in the form of two fractions from a gradient loaded with the product of a blank RNA extraction
(reagents only) was sequenced, and 5 non-template controls (NTC) PCR reactions were amplified with 28-32 cycles.
A mock community (ZymoBIOMICS Microbial Community DNA Standard II; Zymo Research) was also amplified
and sequenced. Library construction and sequencing were performed at the University of Illinois, Chicago DNA Ser-
vices Facility, using an Illumina MiniSeq sequencer (Illumina) in the 2 × 150 cycle configuration (V2 reagent kit).



### 2.4.4 Sequence data processing and detection of labelled ASVs

Primer regions were trimmed off the amplicon sequence data using cutadapt (V2.3, Martin, 2011). All downstream analyses were done in R (V4.0.3 R Core Team, 2020). Quality-trimming and clustering into amplicon sequence variants (ASVs) were done using the DADA2 pipeline (Callahan et al., 2016) with the following quality filtering op-
tions: no truncate, $maxN = 0$, $maxEE = c(2, 2)$ and $truncQ = 2$. Chimaera sequences were removed with removeBimeraDenovo() using the "consensus" method and "allowOneOff". Taxonomic classification of the ASVs was done using assignTaxonomy() against the SILVA database (Ref NR 99; V138.1; Quast et al., 2013). Potential contaminant ASVs were removed using decontam (Davis et al., 2017), employing the default options. Unclassified taxa or those classified as either "Eukaryota", "Chloroplast", or "Mitochondria", and ASVs with a prevalence of
<10% of the samples were removed. The remaining sequences were aligned using SINA (Pruesse et al., 2012) against the SILVA database, and a maximum-likelihood tree was calculated using IQ-TREE (Minh et al., 2020) with an automatic model selection and using the 'fast' option. Beta diversity was calculated with a constrained analysis of principal coordinates (CAP; Anderson and Willis, 2003) from a Morisita-Horn distance matrix of the samples (gradient fractions). Labelled ASVs were detected using differential abundance modelling as described by Angel et al.
(2017). Briefly, for each gradient, the abundance of each ASV in fractions >1.795 g ml-1 (AKA 'heavy' fractions) was compared to that found in fractions <1.795 g ml-1 (AKA 'light' fractions) using package DESeq2 (Love et al., 2014). The Wald hypothesis test and local fit were used, followed by an adaptive shrinkage estimator of the log-fold changes (package 'ashr'). Those ASVs with a statistically significant differential abundance with a P-value < 0.05 and a log2-fold change of 0.26 (ca. 20% average difference in abundance) were considered labelled. Nearest taxon
index (NTI) values were calculated on phylogenetic trees comprised only of the ASVs indicated as labelled by DESeq2 from the 'heavy' fractions of the labelled (non-control) gradients. NTI was calculated as described in (Webb et al., 2002) using the function ses.mntd() in the R package picante (Kembel et al., 2010).2-fold change of 0.26 (ca. 20% average difference in abundance) were considered labelled. Nearest taxon index (NTI) values were calculated on phylogenetic trees comprised only of the ASVs indicated as labelled by DESeq2 from the 'heavy' fractions of the
labelled (non-control) gradients. NTI was calculated as described in (Webb et al., 2002) using the function ses.mntd() in the R package picante (Kembel et al., 2010).

### 2.4.5 Isotopic tracer calculations and statistical modelling

A full list of expressions and formulas used for the following calculations can be found in Table S1. Briefly, isotopic fractions of the different C-pools ($C_{MB}$, WEC, $CO_2$) were calculated by multiplying the at% $^{13}C$ by the concentration
of the C-pool. The amount of C derived from glucose in the measured pools was calculated by multiplying the pool C concentration by the tracer fraction (F; molar ratio of tracer C to total C in the glucose). The amount of glucose-derived WEC in the total WEC (i.e. after correction for $C_{gluc}$ that remained untransformed in the soil, measured enzymatically) was considered microbial extractable exudates ($C_{exud}$). The glucose consumption over time was modelled using a linear model on log-transformed data (to linearise the decrease in $C_{gluc}$ over time). The effect of the
treatment on the microbial biomass, irrespective of time, was modelled using generalised mixed-effect models, with time as a random variable. The function glmer() in the R package lme4 (V1.1-30; Bates et al., 2015) was used with Gamma distribution and the log link function. The turnover rates (i.e., decay) of glucose-derived $C_{MB}$ ($C_{MB-gluc}$) were calculated as the linear regression slope between time and the natural logarithm of $C_{MB-gluc}$. The standard deviation



was estimated by shuffling the replicates and refitting over 999 trials. The accumulation of $CO_2$ against time was modelled using linear regression after inverting both variables (i.e. Lineweaver–Burk plot). Carbon use, storage and stabilisation efficiencies were calculated after 24, 48, 72 and 216 h for oxic and anoxic treatment. Apparent carbon use efficiency ($CUE_A$) assumes the production of microbial biomass only:

$$CUE_A = \left( \frac{C_{MB-gluc}}{C_{gluc} \; uptake} \right)_{(t=72 \; or \; 216 \; h)}$$

Carbon use efficiency (CUE) assumes microbial biomass growth and exudation of water-extractable compounds ($C_{exud}$):

$$CUE = \left( \frac{C_{MB-gluc} + C_{exud}}{C_{gluc} \; uptake} \right)_{(t=72 \; or \; t=216)}$$

The $CUE_A$ and CUE after 24 and 38 hours of incubation, when the concentration of residual glucose under anoxic conditions was still high, were calculated according to Geyer et al. (2019) (Table S1).

Carbon storage efficiency (CSE) characterises biotic and abiotic C accumulation (biomass and both water-extractable and nonextractable transformed C compounds). CSE was calculated from $CO_2$ derived from glucose ($CO_{2-gluc}$) and glucose uptake:

$$CSE = 1 - \left( \frac{CO_{2-gluc}}{C_{gluc} \; uptake} \right)_{(t=72 \; or \; t=216)}$$

Carbon stabilisation efficiency (TCSE) represents abiotic C accumulation in soil, so the proportion of C from glucose that was not converted to $CO_2$ and could not be extracted as $C_{MB}$ or $C_{exud}$:

$$TCSE = CSE - CUE$$

To model WEC emerging from the labelled glucose, the remaining (mean) glucose concentration in soil (enzymatically measured) was first subtracted from WEC. In the statistical model, each observation is weighted by the inverse of the root square of standard deviation calculated as:

## 3. Results

### 3.1 Soil characteristics

At the beginning of the oxic incubation, i.e. after preincubation, the PL soil samples exhibited higher $C_{tot}$, $C_{MB}$ and respiration rates but similar WEC and TN contents compared to the CT soils (Table 1). As expected, the CT samples differed from PL in higher bioavailable iron ($Fe_{avail}$) and $NO_3^-$ content. The brief anoxic preincubation period removed most of the $NO_3^-$, decreased $C_{MB}$ and respiration, and increased WEC and TN but left $Fe_{avail}$ effectively unchanged.




310

*Table 1. Soil characteristics after preincubation, before glucose addition.*

| Site | Incubation | $C_{tot}$ | WEC | TN | $Fe^{2+}$(avail) | $NO_3^-$ | $C_{MB}$ | CO₂ production |
|------|-----------|-----------|-----|-----|------------------|----------|----------|---------------|
|      |           | %         |     |     | $\mu$mol g$^{-1}$ |          |          | $\mu$mol g$^{-1}$h$^{-1}$ |
| PL   | Oxic      | 49.5      | 38.20 | 4.91 | 0.80 | 0.06 | 190.2 | 0.66 |
| CT   | Oxic      | 19.5      | 39.00 | 4.09 | 2.50 | 0.70 | 139.2 | 0.53 |
| PL   | Anoxic    | 48.1      | 118.5 | 6.39 | 0.89 | 0.01 | 146.8 | 0.20 |
| CT   | Anoxic    | 18.4      | 150.8 | 6.02 | 2.75 | 0.01 | 119.5 | 0.20 |

### 3.2 Glucose consumption and consequent changes in bulk C pools

The oxic and anoxic conditions were maintained throughout the incubations. Oxygen levels in the headspace re-
315 mained above 14% in the oxic incubations and were below the detection limit in the anoxic incubations (Fig. S1).
Methane did not accumulate in the incubations and remained at atmospheric levels (data not shown).

Under oxic conditions, glucose consumption was fast. It was removed to below detection limits within 24 hours in
the PL and within 48 h in the CT soils (Fig. 1A). In contrast, a significant amount of glucose was still present after
48 and 216 h under anoxic conditions. Only 56 % (PL) and 70 % (CT) of added glucose were microbially trans-
formed, mainly into water-extractable products of anaerobic metabolism, which corresponded to an increased con-
centration of organic acids in pore water (see below). In both cases, glucose consumption was non-linear, with no ef-
fect of the soil type on the consumption rate (P=0.1). However, not all the C added as glucose was recovered as
newly produced $CO_2$, WEC, $C_{MB}$, or $C_{gluc}$ (Fig. 1 A). Carbon recovered in these pools ranged between 41.5–47.5%
and 73.1–82.6% for oxic CT and PL soils, respectively, and 53.3–99.8% and 59.4–99.4% for anoxic CT and PL
soils, respectively.

The total $CO_2$ (from SOC and glucose) accumulated over time was highest under oxic conditions. In PL, $CO_2$ accu-
mulated up to 66 $\mu$mol C-$CO_2$ g$^{-1}$, while CT samples produced significantly less (up to 45 $\mu$mol C-$CO_2$ g-1; P =
0.04), but both soils produced only 13 $\mu$mol C-$CO_2$ g$^{-1}$ under anoxic conditions, with no difference between the sites
(Fig. 1 C). Similarly, the $^{13}CO_2$ fraction increased with time (Fig. 1 A). However, in contrast to the trends in total
$CO_2$, the $^{13}CO_2$ fraction was similar between the two soils and accounted for approx. 29% of the added labelled gluc-
ose C at the end of the incubation, under oxic conditions, with no statistical significance between the sites (Fig. 1A).
Unsurprisingly, under anoxic conditions, $^{13}CO_2$ reached only 2.5 and 2.9% of the added glucose C in CT and PL
soils, respectively.

Regarding alternative terminal electron acceptors, $NO_3^-$ was low in both soils (Table S3). Under anoxic conditions, it
was close to the detection limit and did not change throughout the incubation, while under oxic conditions, a deple-
tion was observed over time in both soils. In contrast, Fe(II) values remained stable under oxic conditions but in-
creased under anoxic conditions. This was especially pronounced in the Fe-rich CT soil where Fe(II) concentration



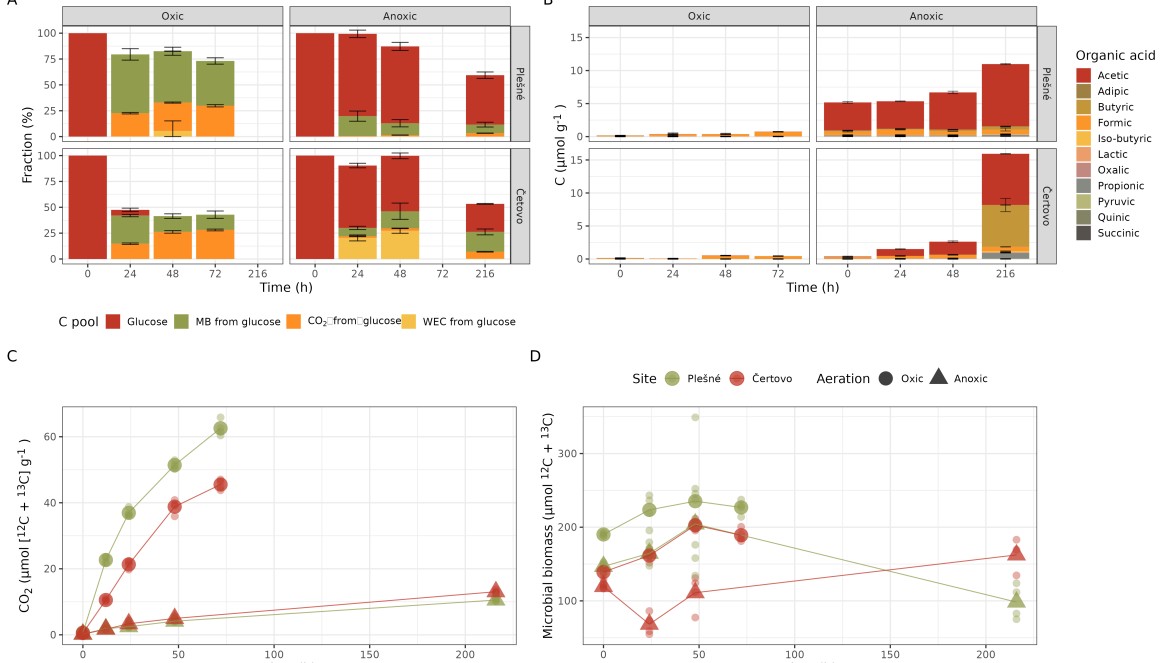

increased nearly 5-fold, indicating vigorous activity of iron-reducing microorganisms. Other potential e- acceptors, such as $SO_4^{2-}$ had deficient concentrations and likely played an insignificant role. Indeed, much of the consumed car-
bon was fermented and released into the soil under anoxic conditions. This can be seen by the accumulation of $^{13}C$ in the form of water-extractable carbon (Fig. 1 A) and the accumulation of OA in pore water, primarily acetate (Fig. 1 B; Table S5). Both WEC and OA were one to two orders of magnitude higher under anoxic than oxic conditions (Fig. 1 A and B). Production of total OA in pore water increased up to 11 and 17 µmol C $g^{-1}$ over 216 h for PL and CT soils, respectively. Acetate accounted for 46-88% of all OA, with the remaining mainly consisting of butyrate,
propionate and formate. Consistent with the low partitioning of glucose C into WEC under oxic conditions (Fig. 1 A), the production of OA was also low, amounting to only 0.58 µmol OA C $g^{-1}$ in PL and 0.28 µmol OA C $g^{-1}$ in the CT pore water, with formate being the most dominant acid (Fig. 1 B). Under oxic conditions, WEC produced from glucose was higher in PL soil than in CT soil. However, under anoxic conditions, it was vice versa, as suggested by a significant interaction of aeration status and soil identity (P < 0.001).oxic conditions, WEC produced from glucose
was higher in PL soil than in CT soil. However, under anoxic conditions, it was vice versa, as suggested by a signi-ficant interaction of aeration status and soil identity (P < 0.001).

*Fig 1. A. Glucose partitioning into measured C pools in the PL and CT soils under oxic and anoxic conditions dur-ing the incubation (means ± propagated errors SEM (see methods); n = 4). $C_{MB}$ refers to microbial biomass meas-ured using chloroform fumigation, and WEC refers to water-extractable carbon. B. OA accumulated during the in-*
*cubation, expressed in µmol OA C $g^{-1}$ (means ± SEM; n=2). C. total $CO_2$ accumulation in the headspace (means ± SEM; n = 4) and D. total microbial biomass (means ± SEM; n = 4).*



### 3.3 Changes in microbial biomass in response to glucose amendments

With glucose amendment, the total microbial biomass ($C_{MB} + C_{MB-gluc}$) increased under oxic conditions by 20% and 36% on average, in PL and CT, respectively, and was consistently higher in PL compared to CT (Fig. 1D; P = 0.02). $C_{MB}$ was consistently lower under anoxic conditions than in oxic conditions (P < 0.01). Microbial biomass derived from glucose ($C_{MB-gluc}$) increased rapidly during the first 24 h in both soils under oxic conditions, but then microbial growth ceased, and $C_{MB-gluc}$ slowly decreased. Similar results were seen in the PL soil under anoxic conditions, although the initial increase in biomass was significantly lower (P < 0.01), and microbial growth ceased even though the added glucose had not yet been consumed. In the anoxic CT soil, biomass increased slowly till the end of the experiment and reached a higher value (19.2 µmol C g$^{-1}$) than the maximum in the anoxic PL soil (11.5 µmol C g$^{-1}$). Surprisingly, only PL exhibited significantly higher newly synthesised biomass under oxic conditions (Fig. 1A; P < 0.01 for the interaction) compared to anoxic conditions.

### 3.4 Microbial carbon use efficiency and turnover rate

Microbial $CUE_A$ and CUE were estimated after 24, 48 h and at the end of incubations (72 or 216 h). However, the calculations after 24 and 48 h were burdened with a large error for the samples under anoxic conditions due to the high concentration of residual glucose in the soil, and the results should be evaluated with caution. They are provided in the supplementary material (Fig. S2). The estimates of $CUE_A$, CUE, CSE and TCSE indicate the potential of the microbiota to transform glucose into different soil carbon pools under oxic and anoxic conditions (Table S1). $CUE_A$, which acknowledges biomass production, was higher in the oxic PL soil than in the oxic CT soil and anoxic PL and CT soils throughout the entire incubation. $CUE_A$ in the anoxic PL soil was higher after 24 h but lower at the end of incubations compared to the anoxic CT soil (Table 2; Fig. S2). $CUE_A$ in the anoxic CT soil exceeded that of oxic CT soil at the end of incubation. CUE, which assumes biomass production and extracellular compounds release, was consistently higher under anoxic conditions as compared to oxic conditions (P < 0.001; Fig. 1). CUE and $CUE_A$ were similar in oxic conditions in both soils, but CUE was significantly higher than $CUE_A$ in anoxic conditions. CUE was higher in PL than in CT soil under oxic conditions throughout the incubation but only after 216 h under anoxia (Table 2; Fig. S2). Carbon storage efficiency, CSE (total proportion of transformed $C_{gluc}$ in the soil at the end of incubation), was much higher under anoxic than oxic conditions (P < 0.001). Anoxic CSE exceeding 0,95 in both soils did not change during incubation, while it decreased in oxic conditions from 0,78 to 0,7 in the PL and from 0,83 to 0,69 in the CT. Carbon stabilisation efficiency—TCSE (proportion of transformed $C_{gluc}$ in a nonextractable pool of SOC)—was difficult to calculate until substantial microbial glucose transformation began, and unused glucose substantially decreased. As a result, TCSE estimates in the anoxic soils were much higher than 1 (from 1.5 to 1.9) after 24 and 48 hours of incubation. TCSE at the end of the experiment was higher in Fe-rich CT soil under both oxic and anoxic conditions (Table 2). In the oxic conditions, TCSE increased during incubation from 0.21 to 0.27 in the PL soil and from 0.48 to 0.53 in the CT soil. The TCSE was significantly higher than that of anoxic soils in both cases. Regarding microbial turnover, $C_{MB-gluc}$ in the CT oxic soils exhibited twice as much turnover as $C_{MB-gluc}$ in the oxic PL soil (Table 2). Under anoxic conditions, biomass turnover was very low in the PL soil; however, it cannot be calculated in the CT soil as microbial biomass was still growing after 216 h. at the end of incubation. CUE that assumes biomass production and extracellular compounds release, was consistently higher under anoxic conditions as compared to oxic conditions (P < 0.001; Fig. 1). CUE and $CUE_A$ were similar in oxic conditions in both soils, but CUE was significantly higher than $CUE_A$ in anoxic conditions. CUE was higher in PL than in CT soil un-



der oxic conditions throughout the incubation but only after 216 h under anoxia (Table 2; Fig. S2). Carbon storage efficiency, CSE (total proportion of transformed $C_{gluc}$ in the soil at the end of incubation), was much higher under anoxic than oxic conditions (P < 0.001). Anoxic CSE exceeding 0,95 in both soils did not change during incubation, while it decreased in oxic conditions from 0,78 to 0,7 in the PL and from 0,83 to 0,69 in the CT. carbon stabilisation

efficiency, TCSE (proportion of transformed $C_{gluc}$ in a nonextractable pool of SOC) was difficult to calculate until substantial microbial glucose transformation began, and unused glucose substantially decreased. As a result, TCSE estimates in the anoxic soils were much higher than 1 (from 1.5 to 1.9) after 24 and 48 hours of incubation. TCSE at the end of experiment was higher in Fe-rich CT soil under both the oxic and anoxic conditions (Table 2). In the oxic conditions, TCSE increased during incubation from 0.21 to 0.27 in the PL soil and from 0.48 to 0.53 in the CT soil.

The TCSE was significantly higher as compared to anoxic soils in both cases. Regarding microbial turnover, $C_{MB\text{-}gluc}$ in the CT oxic soils exhibited twice as much turnover as $C_{MB\text{-}gluc}$ in the oxic PL soil (Table 2). Under anoxic conditions, biomass turnover was very low in the PL soil; however, it cannot be calculated in the CT soil as microbial biomass was still growing after 216 h.

**Table 2.** *Mean values of Carbon use efficiencies ($CUE_A$ excluding exudates and CUE including exudates), soil car-*
*bon storage (CSE) and stabilisation (and TCSE) efficiencies, derived from glucose, with confidence intervals in the parentheses at the end of incubation, and mean microbial turnover of newly formed biomass (±standard deviation). Different letters above numbers denote significant differences for values in each column.*

| Site | Incubation | $CUE_A$ | CUE | CSE | TCSE | Biomass turnover (d-1) |
|---|---|---|---|---|---|---|
| PL | Oxic | 0.43[b] (0.39-0.47) | 0.43[a] (0.40-0.46) | 0.70[a] (0.69-0.71) | 0.27[a] (0.25-0.30) | 0.136[a] ± 0.015 |
| CT | Oxic | 0.16[a] (0.13-0.20) | 0.16[c] (0.14-0.19) | 0.69[a] (0.69-0.70) | 0.53[c] (0.41-0.69) | 0.320[b] ± 0.039 |
| PL | Anoxic | 0.14[a] (0.11-0.17) | 0.83[b] (0.71-0.97) | 0.95[b] (0.93-0.96) | 0.12[b] (0.09-0.15) | 0.084[c] ± 0.014 |
| CT | Anoxic | 0.3[c] (0.21-0.43) | 0.72[d] (0.656-0.92) | 0.90[b] (0.88-0.91) | 0.17[d] (0.12-0.27) | n.d. |

n.d. Not determined.

### 3.5 Incorporation of $^{13}C$ into nucleic acids

Sequencing produced 17M reads with 49K ± 22K per sample. Quality filtering, merging and chimera filtering removed 20% of the reads (22% ± 9% per sample; Table S6). In addition, 60 ASVs were flagged as potential contaminants and were removed. Taxonomy-based filtering steps removed an additional 274 OTUs (0.156% of the sequences) classified as chloroplast and mitochondria or did not classify at the kingdom level. Taxonomic orders whose ASVs had a cumulative prevalence of <5% (30 samples) were also removed (100 ASVs, 0.046% of the

reads), leaving 14,974 taxonomically assigned ASVs (Table S7). Lastly, since RNA-SIP assumes that all ASVs should be present in all fractions of an individual gradient, rare reads appearing in <10% were removed (10,406



ASVs, 2.37% of the reads). Beta diversity analysis showed a significant deviation of the community in the heavy fractions from the light fractions of the density gradients under oxic conditions, indicating the labelling of a sizable fraction of the microbial community. Some separation was also observed in the samples incubated under anoxic con-

ditions, though to a much smaller extent, indicating that only a minority of the community was labelled. As expected, no significant deviation was seen for the unlabelled (control) samples (Fig. S3).

Differential abundance modelling using DESeq2 identified, in total, 330 unique ASVs as significantly more abundant in the 'heavy' fractions compared to the 'light' fraction (Table S8, Fig. 2, Fig. S5–S8). Only a small minority of those (21 in the oxic and 0 in the anoxic incubations) were detected in the unlabelled gradients and are regarded as

false positives. As expected, most of the labelled ASVs were found in the oxic incubations, with 192 and 149 unique ASVs labelled across all samples from CT and PL sites, respectively, making up 28% and 32% of all the reads in these samples (138 ASVs were mutual). Most of these OTUs belonged to the dominant phyla: Proteobacteria, Actinobacteriota, Acidobacteriota, Verrucomicrobiota, and Bacteroidota. Other prevalent phyla were Planctomycetota, Crenarchaeota, Candidate Phylum RCP2-54, Candidatus Palusbacterota (WPS-2) and Myxococcota. Under anoxic

conditions, only 48 and 38 ASVs from CT and PL sites were labelled, making up only 4% and 6% of the reads in these samples, respectively. These ASVs mainly belonged to the Firmicutes.

**Fig 2. Identifying labelled ASVs using differential abundance analysis.**

*Fold change (log$_2$) of ASVs between fractions where $^{13}$C-labelled RNA is expected to be found (>1.795 g ml$^{-1}$; AKA 'heavy' fractions) to the fractions where unlabelled RNA is expected to be found (<1.795 g ml$^{-1}$ AKA 'light' frac-*

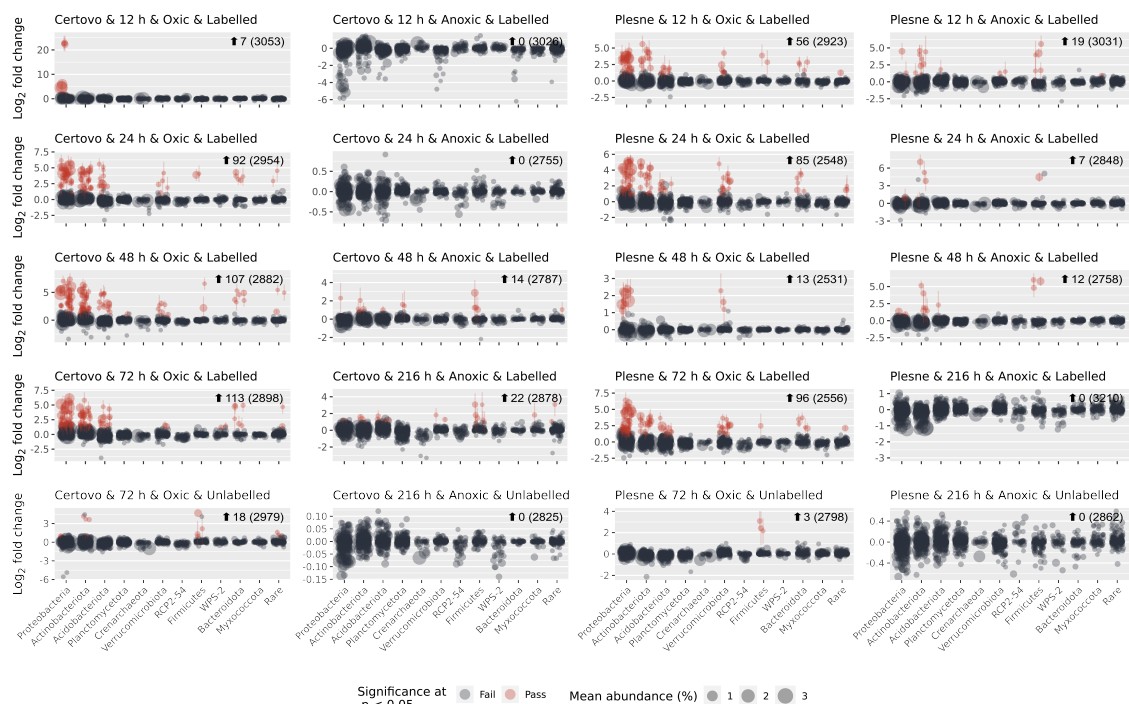

*tions). Each circle represents an ASV, which passed sparsity filtering (see Materials and Methods). The X-axis*



*shows the taxonomic phylum-level classification of the ASVs (ordered by total abundance across all samples), while the Y-axis is the mean log$_2$ fold change across all gradients. The size of each circle represents the normalised abundance across all samples. Red circles denote ASVs flagged as significantly more abundant in the heavy fractions than the light fractions, with a log$_2$ fold change of >0.26 (~20% increase) and an adjusted P-value of <0.05.*

Under oxic conditions, labelling of ASVs was relatively rapid and incubations past 24 h showed little change in the number and identity of the labelled ASVs (Fig. 2). Of the two sites, PL was quicker to respond and displayed already labelled ASVs in all major phyla after 12 h, while the Fe-rich site—CT–had only a few Gammaproteobacteria ASVs labelled at this point. PL was also quick to respond under anoxic conditions and displayed several labelled ASVs from order Corynebacteriales (Actinobacteria) and the class Bacilli already after 12 h. On the other hand, nearly all

labelled ASVs from CT under anoxic conditions first appeared after 48 h, or even 216 h, of incubation.

The labelled ASVs showed not only consistency over time (i.e., the same ASVs labelled at different time points) but also a significant phylogenetic clustering. Measuring the nearest taxon index (NTI, i.e. the standardised measure of the phylogenetic distance to the nearest taxon for each taxon in the sample) for the labelled ASVs showed that the values averaged around 1.5 standard deviations across all the SIP gradients (Fig. S4). Interestingly, in most taxo-

nomic groups, either identical or very closely related ASVs were labelled in the samples from both sites (Fig. 3). However, some interesting exceptions are noted. Under oxic conditions, CT samples had more labelled ASVs from the order Frankiales (Actinobacteria) and Xanthomonadales (Gammaproteobacteria), while Plešné had more labelled ASVs affiliated with the class Verrucomicrobiae. Under anoxic conditions, only Plešné had some labelled ASVs from the orders Corybacteriales (Actinobacteria) and Acidobacteriae.







**Fig 3. Phylogenetic trees of ASVs that passed the sparsity threshold.** Coloured dots on the tree tips designate either the taxonomic order (for the classes Actinobacteria, Alphaproteobacteria and Gammaproteobacteria) or the class (for the phyla Acidobacteria, Verrucomicrobiota, Bacteroideta and Firmicutes) to which the ASVs were assigned. The associated heatmaps are divided into Plešné (PL) and Čertovo (CT) samples under oxic (Ox) and anoxic (Anox) conditions at different time points. Highlighted tiles in the heatmaps denote ASVs with a statically significant differential abundance (labelled ASVs), and their colour intensity is proportional to their log2 fold change between the 'heavy' and 'light' fractions.

## 4. Discussion

Although CUE has been employed in numerous studies to understand soil C cycling (e.g. Herron et al., 2009; Tao et al., 2023), researchers typically assume fully oxic conditions in the soil and disregard the effects of anoxia, alternative electron acceptors, and the fate of secreted products (Manzoni et al., 2018). This incubation experiment was designed to investigate the differential response of microbial carbon use, carbon storage and stabilisation efficiencies (response variables) to glucose amendments (controlled variable) under varying levels of oxygen and available Fe in the soil (explanatory variables). The amount of added glucose was intentionally small and mimicked the input of available C to meet the energy demand for optimal metabolic activity of the metabolically–activated population until the substrate was depleted (Anderson and Domsch, 1985). The amount of added glucose (approx. 100 µmol C g$^{-1}$) was within a range of bioavailable C in the studied soils (WEC from 38 to 150 µmol C g$^{-1}$) and lower than C in microbial biomass ($C_{MB}$ 120 to 190 µmol C g$^{-1}$) and thus is unlikely to have caused a significant disturbance.

### 4.1 CUE values were greatly affected by the aeration status of the soil

Carbon use, storage and stabilisation efficiencies estimated in our experiments were linked to the ability of the microbial community to assimilate and distribute glucose to catabolism and biosynthesis of microbial compounds, either intra- or extracellular. Accordingly, it should be interpreted as glucose carbon use efficiency in the sense defined by Schimel et al. (2022). The efficiency of microbial metabolism and C retention in the soil was calculated for all sampling points of the experiment (24, 48 and end of incubation after 72 and 216 h, respectively), but under anoxic conditions, the estimates were burdened with large error under conditions of high concentration of unused glucose in the anoxic soil during earlier samplings, similarly to Geyer and coauthors (Geyer et al., 2019).In contrast, glucose consumption was fast under the oxic treatment, and the turnover of microbial biomass was in order of a few days (72 and 168 hours for CT and PL, respectively). Such rapid turnover makes CUE estimates biased by microbial biomass recycling. Since the character of $^{13}$C substrate being metabolised changes once the microbial biomass gets recycled, CUE estimates specifically derived for glucose metabolism cannot be comparable between treatments. Consequently, we will discuss only the results after 72 h and 216 h, respectively, for oxic and anoxic treatment. . We are aware that the interpretation of results obtained at different time intervals might be different. However, as mentioned in the introduction, the measures of soil C use and stabilisation efficiencies are influenced by microbial growth. At the end of the experiment, the growth of the microbial population ceased, apart from the anoxic CT soil, where the microbial biomass was still slowly growing. We believe this approach makes our comparison more reliable than if we were to compare the same time intervals.



The CUE value, especially under oxic conditions, was potentially affected by the rate of microbial turnover. The higher the turnover rate, the lower the CUEs. $CUE_A$ and CUE were almost identical in the oxic soils, showing either negligible production of exudates, re-consumption of microbially transformed C (Geyer et al., 2019), or formation of unquantified mineral-organics associations (MOAs), mainly in the Fe-rich CT soil. Oxic CT soil had lower CUEs than oxic PL soil, presumably because of the high turnover of $C_{MB}$ associated with a community with more copiotrophic members (see below), C stabilisation into MOAs in an Fe-rich environment accompanied with lower accessibility of C for microorganisms, and P limitation (Čapek et al. 2016). The effect of Fe(III) C stabilisation on CUE is highlighted by the high proportion of nonextractable C in this soil.

Under anoxia, the CUE of the soils was three to four times higher than $CUE_A$ and, therefore, even higher than under oxic conditions. The difference is attributed to the high production of fermentation products, mainly organic acids, as is confirmed by their increase in concentration in the pore water. The amount of all microbially transformed C remaining in soil at the end of the experiment (CSE) was higher under anoxia, but its stabilisation (TCSE) was lower. This is related to the solubility and ease of degradation of organic acids (Ström, 1997) and solubility of Fe(II) bounded compounds.

Iron in the CT soils was associated with lower CUE and $CUE_A$ than the PL soil but increased C stabilisation of newly transformed C in both oxic and anoxic conditions. Although this data was obtained from the breakdown of a sole substrate, it confirms that the production of extracellular metabolites must be considered when evaluating the efficiency of microbial metabolism in relation to soil organic matter storage. The data further support that the abiotic formation of MOAs from microbially transformed carbon must not be neglected as it may significantly impact anoxic CUE values.

### 4.2 Unrecoverable C from soil amendments

In our study, much of the added $^{13}C$ could be traced in the $CO_2$, microbial biomass or WEC, while a significant amount remained untraced. Presumably, the $^{13}C$ was stabilised via adsorption on mineral surfaces or co-precipitated, forming MOAs. (i) Small amounts of glucose can be sorbed on Fe(hydroxy-)oxides via H bonds (Olsson et al., 2011) or associated via co-precipitation processes (Lenhardt et al., 2023). (ii) metabolism intermediates can be firmly bound on MOAs (Jones and Edwards, 1998). (iii) Necromass interacts with mineral surfaces to form bioorganic complexes, which are assumed to be the main factors driving soil necromass stabilisation (Camenzind et al., 2023). (iv) Extracellular enzyme preservation can be attributed to copolymerisation, adsorption, and encapsulation (Gotsmy et al., 2021). Lastly, it has been documented that the extractability of proteins from soils ranges from 1 to 5 % (Benndorf et al., 2007; McClaugherty and Linkins, 1988), which could account for some of the untraced parts of the labelled C-pool. Sorption of organics to minerals to form MOAs decreases their decomposability and microbial reuse (Jones and Edwards, 1998; Porras et al., 2018).

### 4.3 Aeration status and Fe availability affected C partitioning into different pools.

Under oxic conditions, a major part of added glucose disappeared from the soil within the first 24 h and, accordingly, $C_{MB}$ and $CO_2$ from glucose increased sharply during this period. However, part of the added C was shifted to the nonextractable C pool (about 20% and 50% in the PL and CT soil, respectively), most likely as a result of the stabilisation effect of MOAs (see above). Glucose incorporation to $C_{MB}$ and respiration was lower but C stabilisation was



higher in the Fe-rich CT soil compared to the Fe-poor PL soil. Slower glucose transformation in the CT soil was pre-
sumably caused by a combined effect of nutrient limitation, mainly by P, in the CT soil but not in PL soil (Tahovská
et al., 2018; Čapek et al. 2016), higher organics binding to minerals to create MOAs, higher organics protection in
aggregates, or by different structure of microbial community with lower metabolic efficiency (see below). A slight
but detectable increase in WEC from glucose and increasing concentration of OAs in the soil solution suggest the
presence of anoxic microniches in oxic soils (Borer et al., 2018), where oxygen-tolerant bacteria carry fermentation
with formic acid being a predominant product. Indeed, the concentration of formic acid increased in the pore water
in both oxic soils by one order of magnitude during the incubation. Acetate and other tricarboxylic acids, concentra-
tions of which increased in the pore water, can also originate from overflow metabolism in conditions of carbon ex-
cess (Basan et al., 2015).

Under anoxic conditions, acetate accumulation suggests that acetogenesis dominated fermentation in both soils,
which has been observed previously for forest soils (Küsel and Drake 1995, Degelmann et al. 2009). However, acet-
ate accumulation in the Fe-rich CT soils was considerably delayed compared to the PL soil. Concomitant with the
drop in $NO_3^-$ values during this incubation period and the presence of Fe, this indicates that alternative processes to
acetogenesis were active as C-sinks in CT soil. This phenomenon was seen in other Fe-rich soils (e.g. Küsel et al.
2002, Lentini et al. 2012).

CT soil contains almost seven times more oxalate-extractable Fe(III) than PL soil. However, only a small fraction of
oxalate-extractable iron is microbially reducible (Lovley and Phillips, 1987). Most iron minerals in soil exist in
forms which are unavailable to microorganisms. Amorphous Fe(III) oxides, e.g. ferrihydrite, are considered the most
microbially accessible forms of Fe(III) (Lovley and Phillips, 1987; Lentini et al., 2012). For PL soil, bioavailable
Fe(III) constitutes roughly 11% of oxalate-extractable Fe(III) in oxic conditions and roughly 13% in anoxic condi-
tions (Table S2). These values for CT soil are approximately 14% and 21%, respectively. This suggests that the on-
set of anoxic conditions in iron-rich CT soil might have triggered reactions resulting in the release of iron solids
from bound structures, thus making iron available for microbial reduction. Also, iron reduction can mobilise nutri-
ents and carbon previously bound to iron minerals. Organic carbon can be additionally released from soil organic
matter (Bhattacharyya et al., 2018). Both soils demonstrated Fe(II) accumulation after 216 hours under anoxic condi-
tions. CT had double the amount of reduced iron compared to PL soil. It has been reported that not only can Fe(III)
serve as a terminal electron acceptor in microbial metabolism, but it also acts as an electron sink for fermentation
(List et al., 2019). While microorganisms gain more energy from iron respiration than fermentation, the main benefit
of iron reduction is buffering pH and extending the acidogenic phase of fermentation, in which organic acids are pro-
duced (Wang et al., 2019). Our findings show that iron soil content significantly impacts microbial metabolism in an-
oxic environments.

### 4.4 A plethora of aerobic and anaerobic bacteria were activated under varying oxic conditions.

The composition of the active part of the microbial community depends on many edaphic factors, among them the
aeration status and the availability and type of Fe minerals and C source present at the site (Lentini et al., 2012; An-
gel and Conrad, 2013; Barnett et al. 2021). Not surprisingly, oxygen availability was selected for certain microbial
taxa in both PL and CT incubations, and many taxonomic groups of well-known strict aerobes were only labelled un-
der oxic conditions. Complete glucose removal under oxic conditions supported the growth (and hence labelling) of
many more ASVs from all phyla compared to anoxic conditions. Much-extended growth under oxic conditions was





expected simply thanks to the higher energy yield from oxic respiration compared to anoxic respiration or fermentation. Active taxa under oxic incubations included Acidobacteriota Subgroup 1 (Eichorst et al., 2018), members of the
Beijerinckiaceae family (Hyphomicrobiales/Rhizobiales; Dedysh and Dunfield, 2016) and the order Acetobacterales (Sievers and Swings, 2015) of the Alphaproteobacteria, members of the orders Xanthomonadales (Saddler and Bradbury, 2015) and Burkholderiales (genus Burkholderia; Garrity et al., 2015). Nearly all members of the abovementioned groups are obligate aerobes. Only a handful of ASVs affiliated with Bacillota (formerly Firmicutes) were labelled in our incubation under oxic and anoxic conditions, even though nearly all members are known to be strict or
facultative anaerobes. These Bacillota (all Lactobacilli) were more numerous and strongly labelled under anoxic conditions. However, the labelling of these facultative anaerobes (Pot et al., 2014) under oxic conditions indicates their activity in hypoxic microniches. The calculated NTI values above 1.5 imply that the labelled taxa are more phylogenetically related than expected by chance (Stegen et al. 2012) and indicate a concerted response of specific metabolic guilds to the amendment and incubation conditions.

Despite the differences in iron content, only minor differences were observed in the taxonomic groups that became labelled under either oxic or anoxic conditions in the samples from the two sites. Samples from the iron-rich soils of CT had nearly all Xanthomonadales ASVs (all Rhodanobacter or Dyella) and Acidothermus (Frankiales, Actinomycetota; formerly Actinobacteria) labelled, while PL had close to none from these taxa. Many Xanthomonadales are known to couple denitrification to ferrous iron oxidation (Huang et al., 2021), and Acidothermus have been associ-
ated with (acidic) denitrifying conditions (Bárta et al., 2017). Similarly, many denitrifying bacteria were shown to be labelled in SIP experiments of soils under high moisture levels and reducing conditions (Greenlon et al., 2022; Coskun et al., 2019). These groups may have been responsible for the nitrate depletion in our incubations. Moreover, the Fe-rich CT soils had lower CUE, correlated with a more active copiotrophic community (whose metabolism is less efficient). Indeed, many of the labelled ASVs in this soil belong to fast-growing, copiotrophic phyla such as Ac-
tinobacteria (Ramirez et al., 2012) and Gammaproteobacteria (Fierer et al., 2012), while the PT soils were differentially more enriched with oligotrophic phyla such as Acidobacteroita (Fierer et al., 2012) and Verrucomicrobiota (Bergmann et al., 2011).

## 5. Conclusions

Our findings imply that, regarding CUE, soil aeration status primarily affects immediate C incorporation into soils, with anoxic soils having slightly higher CUE. C is used mainly under oxic conditions for biomass and $CO_2$ production, while in anoxic conditions, C is used primarily for producing extracellular exudates. High Fe content in the CT soil constrained $CUE_A$ and CUE under oxic conditions but only CUE under anoxia, while $CUE_A$ increased. Under anoxic conditions, exudates remained mainly in the water extractable C pool and less C was lost as $CO_2$, leading to
higher CSE as compared with soils under oxic conditions. This enhanced production of microbial exudates at the expense of microbial biomass under anoxic conditions suggests that soil aeration and mineralogy are important determinants of lateral C export fluxes and may thereby influence C storage in the soil. However, C stabilisation potential, TCSE, remained lower in the anoxic soils, implying biomass C has a higher potential for long-term C sequestration. In contrast, extracellular exudates might have a significant role as a source of available C, which can accelerate the
development of microbial communities when environmental conditions improve (e.g. after aeration of flooded soils).



Fe content promoted the stabilisation potential of the soil (TCSE) under both oxic and anoxic conditions showing an important effect of soil mineralogy.

### Data and code availability

Raw sequences were deposited into GenBank's SRA database under accession no. XXXX. The scripts to reproduce the analyses are available online: https://github.com/ISBB-anoxic/Anoxic_CUE.

### Author contribution

Conceptualisation: HŠ, JN, RA, TBM; Lab work: JN, ACL, SJ; Formal analysis: RA, HŠ, PČ, TBM; Writing - original draft: JN, HŠ, RA, TBM, Writing - review & editing: all authors; Visualisation: RA; Funding acquisition: HŠ.

### Competing interests

The authors declare that they have no conflict of interest.

### Acknowledgements

We thank Jiří Kaňa for the help in sampling. Ljubov Polaková is acknowledged for her tremendous support of stable isotope measurements. Eva Petrová prepared the RNA samples for sequencing.

### Financial support

JN was supported by the EU grant Rozvoj JU – Mezinárodní mobility (CZ.02.2.69/0.0/0.0/16_027/0008364). RA was supported by supported by the MEYS CZ - Operational Programme RDE (SoWa Ecosystem Research; project no. CZ.02.1.01/0.0/0.0/16_013/0001782). ACLR was supported by the MEYS CZ - Operational Programme RDE (SoWa Ecosystem Research; project no. CZ.02.1.01/0.0/0.0/16_013/0001782). HS and PC were supported by the Czech Science Foundation projects 22-05421S and 20-14704Y. TBM and SJ were supported by the Czech Science Foundation (20-22380S). Stable isotope measurements were supported by supported by MEYS CZ - Operational Programme RDE (SoWa Ecosystem Research; project no. CZ.02.1.01/0.0/0.0/16_013/0001782), and the MEYS CZ Large Infrastructure for Research MEYS LM2015075, NSF-OCE 1736656 (LIA).



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
