# Peer review of "Aeration and mineral composition of soil mediate microbial CUE"

_EGUsphere, 2025_

## Author Response (AR1)

**Editor**

I've seen the reviewer comments and authors' response on how CUE is measured and what components are included. I sense that there a little confusion of what CUE actually means. CUE is a physiological trait of microorganisms often measured in soil as a community aggregate or emergent trait. It determines how the carbon taken up is used in anabolic and catabolic pathways. It is an intrinsic property of micro-organisms. What happens to the secreted products or microbial cell materials after death is not to be considered in CUE calculation. If we want to measure the balance of C remaining in soil to that lost to the atmosphere we need a different index, that is not CUE. Please consider this while the revisions are made and submitted.

Dear Ashish Malik,

Thank you very much for handling our manuscript and for your comment regarding the interpretation of CUE. We revised the ms and clarified our operative definition of CUE in lines 11-21, 43-52 and 64-71.

**Reviewer 1**

Review of EGUSPHERE-2025-481

Aeration and mineral composition of soil determine microbial CUE

The authors report on a study of substrate-carbon allocation in soil microbial communities, depending on aerobic respiration versus anaerobic metabolism (iron respiration, fermentation), using two soils differing in Fe content, and performing 13C-glucose tracing measurements under aerobic and anoxic conditions. Going beyond just measuring 13C in microbial biomass and in respiration the authors also tried to measure substrate consumption (glucose depletion) and exometabolites secretion (organic acid accumulation). Using these data the authors calculate different proxies of microbial carbon use efficiency i.e. ambient CUEa, CUE, CSE, and TCSE. And they coupled these to the measurement of active microbes incorporating 13C-glucose in their RNA by RNA-SIP. A very nice move.

My main issue with the current paper is the following ( would suggest to discuss this at least in a revised version of this manuscript, if the authors do not confer):In their C allocation scheme the authors 'mix' low- and high-molecular weight (LMW and HMW) metabolites into one class, exudates or extracellular metabolites (see Line 31-49). However, I would criticize this strongly as polymeric forms of exudates such as exopolysaccharides and exoproteins likely contribute to non-growth anabolic production, accumulating in soil through sorption (they would need to be added to 13C in biomass or 13C-growth but they were not measured here) while LMW exometabolites either do almost not sorb and therefore do not or negligibly contribute to SOC formation (sugars), or rather mobilize SOC and desorb it (organic acids) or only some/slightly sorb (amino acids, pH dependent). While HMW-C exudates therefore clearly are anabolic outlets of microbial metabolism and contribute to SOC formation together with microbial growth and biomass formation, LMW-C exudates - if formed by fermentation - are catabolites (acetate, formate, butyrate) and should therefore rather be classified as such (catabolites) to be summed with CO2 and CH4. However, adding catabolites to the anabolic processes and to the growth plus non-growth anabolic component does not make sense here. Organic acids per se likely do not contribute to SOC formation but rather decrease it by mobilizing and solubilizing SOC. To my mind, therefore, the exometabolites that the authors measured (organic acids) should not be added to the growth side but to the catabolic side. Unless, under changing conditions those fermentation products which can constitute a major fraction of consumed sugar (glucose) under anaerobic conditions to recycle NAD+ from NADH+, but under oxic conditions they might be used by aerobic microbes and utilized for growth and energy formation (mentioned finally in Lines 609-612). But this is a different situation than tested and discussed here, either anoxic or oxic, no switches tested. Acetate and formate – if further metabolized under anoxic conditions – are largely ending up in CH4 and CO2 and little in biomass, so they remain catabolites.

My overall conclusion on this is therefore that just measuring 13C-respiration and 13C-microbial biomass as many researchers have done under aerobic conditions and summing them for microbial uptake to estimate CUE as biomass/uptake would underestimate total uptake and therefore overestimate CUE under anaerobic/fermenting conditions. Adding organic acid formation as catabolites would increase microbial uptake, would not affect growth estimates and overall considering fermentation products would decrease CUE (CUEa to CUE conversion) estimates under anaerobic conditions. Beautifully in this work the authors assessed uptake not by summing respiration and growth (and other products such as exometabolites) but by measuring the depletion of added substrate (glucose). So, in this paper uptake would not have changed considering exometabolites or not, and growth efficiency as growth/uptake would not be affected by any of those considerations. In contrast to this, including extracellular polymeric outputs (EPS-protein and -sugars) not related to growth

would increase anabolic outputs as a fraction of C uptake and therefore increase CUE, under both aerobic and anaerobic conditions.

Thank you very much for the thorough review of our work!

The goal of our study was to show the fate of consumed glucose in the soil under oxic x anoxic conditions and to show that one should be careful using CUEa (counted from biomass and respiration only).

We are aware of the issue of mixing LMW and HMW in the Introductory part. We should have clearly defined products of energy metatolism and biosynthesis. We define different products of energy metabolism and biosynthesis in revised version of Introduction from line 30 to line 70 .

It would be nice to distinguish between anabolic and catabolic metabolites but it is not simply possible separating HMW and LMW without more detailed (qualitative) analyses that are beyond the scope of our study.

We agree that in terms of microbial physiology substances derived from catabolic processes should be assigned to $CO_2$ and those associated with anabolism to biomass. However, the MS is not focused directly on microbial physiology and did not have ambitions to study distribution of consumed C into catabolic and anabolic processes in different environmental conditions. We wanted to show that CUE, frequently used in ecosystem studies (a rather mechanistic approach) to estimate C balance and C storage in soils, can be biased when organic compounds released from cells are neglected. Considered from this perspective, connecting catabolic extracellular compounds with $CO_2$ is technically correct physiological approach. Nevertheless, these extracellular organic compounds can be quickly reused by microbes and can be build back into biomass under any conditions. In that situation, one may wish to add these compounds in the denominator (to glucose) or nominator (to biomass). In addition, part of released compounds can be bounded on organomineral complexes and remains in the soil. Due to such an ambiguity, there is no single correct approach of CUE calculation. It always depends on the specific research question. We believe that to answer our main question (i.e. how much is CUE (in the mechanistic sense) distorted in different conditions (oxic x anoxic x Fe content), if it is calculated only from microbial biomass and $CO_2$ and how it affects C balance in soil),  adding organic catabolites to „anabolic" part is reasonable.

We agree that organic acids can solubilize OM – the main mechanism is acidification (this was not the case in our experiments – pH did not decrease). On the other hand, we do not agree that  LMW compounds cannot be stabilized. They can be stabilized  by organo-mineral association (association with Fe and Al hydroxyoxides) and binding can be fast and strong (e.g. Jones, D. L., & Brassington, D. S. (1998).. European Journal of Soil Science, 49(3), 447-455., Dippold, M., Biryukov, M., & Kuzyakov, Y. (2014). Soil Biology and Biochemistry, 72, 180-192.). Iron hydroxyoxides are one of the strongest sorbents. One should also consider the fact that the studied soils are already acidic (pH 3.5) and hence solubilisation of SOC because of organic acid release from fermentation is unlikely. The main text was rewritten according to the comments (details below).

More detailed comments:

Line 11: some sentences are unclear, imprecise and might be improved, e.g. this one “Microbial carbon use efficiency (CUE) in soils is used to estimate the balance of $CO_2$ respired by heterotrophs versus the accumulation of organic carbon (C).”

We revised this sentence to "Microbial carbon use efficiency (CUE) is used to estimate the proportion of organic substrate (glucose) consumed by microbial biomass that is not released from soil as CO2, i.e. remains in the soil as some anabolic product."

Line 13: biomass growth was not assessed here, only 13C immobilization (uptake and retention) in microbial biomass, which can be stored or used in cell size and cell number growth.

We reformulated the sentence: "We estimated CUE based on measured cumulative microbial respiration, residual glucose , biomass , and extracellular metabolites concentration."

Line 18-19: due to the reasons discussed in the paragraph above I do not copy the conclusion in the abstract, neither the "short-term C preservation" by exudation f organic acids (!), nor the "underestimation of apparent CUE" - "Our findings confirm that anoxia in soils enhances short-term C preservation. Accordingly, excluding exudates in mass flux calculations would underestimate apparent CUE values."

We revised this sentence to "Our findings confirm that anoxia in soils enhances short-term C preservation. Accordingly, excluding exudates in mass flux calculations would underestimate C retention in the soil."

Line 24: rewrite "soil C storage"

"Storage" has been replaced with "retention"

Line 64: EPS is mostly polysaccharides and polypeptides (instead of amino acids)

The sentence has been revised to "prokaryotes secrete extracellular polymeric substances (EPS; mostly polysaccharides and polypeptides) and amino acids to limit water loss" (L 77-78)

Line 69-70: this sentence is completely unclear….

Lines 67-70 were removed

Line 111-112: Fe(tot) is given in mM kg-1 i.e. mmol per liter of volume (usually solution) per kg dry soil, I guess this should be mmol kg-1?

Corrected here and in Table S2 to mmol kg-1

Line 294, 298: CSE (carbon storage efficiency) – this largely comprises 13C incorporated in **microbial organic C products** (including biomass, necromass, and secreted products), TCSE (carbon stabilization efficiency) – this largely comprises 13C incorporated into **microbial necromass**, excluding biomass and secreted products. If both are true this might be additionally mentioned to make it easier for the readers to follow.

The description was revised see lines 295-315.

Line 368+: In part the paper is super detailed e.g. in section 3.4 which might be shortened for readability.

This part was shortened.

Line 386: you mention that in anoxic soils TCSE was larger than 1, but from the equation used to calculate this CSE-CUE this is simply impossible. Please check.

1. Thank you for finding this error.  TCSE was indeed lower than one, it was even negative as (i) CUE calculus was biased by high amount glucose remaining in the soil and (ii) part of consumed glucose could remain unchanged in cell for later use and detected in biomass pool (Bremer, E., & Kuikman, P. (1994). Soil Biology and Biochemistry, 26(4), 511-517; Picek, T., Šimek, M., Šantrůčková, H. 2000. Biology and Fertility of Soils 31, 315-322.)

Line 505: the reason that CUE is higher in anoxic than in oxic conditions is that catabolite secretion is counted to growth and anabolism instead, which increases CUE. See my critique previously.

See our explanation (defence of our approach) above

Line 517-528: this is a simply list of potential 13C sinks. My "feeling" says that a large fraction of the unaccounted, non-extractable 13C ion the soils is in microbial necromass, given the fast metabolism and turnover of microbial biomass. Fast microbial turnover (70-170 hrs) means that a large part of that biomass 13C in polymeric forms is transformed into necromass, remaining adsorbed, unextractable, HMW. Having microbial turnover and microbial biomass would even allow to calculate the fraction of 13C in necromass.

Yes, you are right - we wanted to show possible mechanisms, but there is no sentence in the MS that states which of the mechanisms is important. We have added it: We hypothesize that necromass plays the most important role, along with the binding of organic compounds in organo-mineral complexes. This is suggested by the higher TCSE in CT, an Fe-rich soil, and the rapid biomass turnover in this soil under oxic conditions.

Line 541: acetate is not a tricarboxylic acid but a di-

Corrected

Line 572: please rephrase "much-extended growth" which I do not grasp in terms of meaning.

The sentence was revised to "The much higher RNA-labelling under oxic conditions was expected "

Line 593: though it has been discussed or claimed that copiotrophs are less energy and carbon efficient that oligotrophs I have rarely seen convincing data for this. Actually copiotrophs which grow fast should waste less energy in respiration, have lower maintenance respiration, have less costs for exoenzyme production and therefore allocate more anabolic output into growth, and therefore have higher CUE. Moreover their genomes are often more simple and smaller, consuming less resources in replication.

This is a fascinating topic which is still developing. Still, from theory (and several experimental evidences) copiotrophs are *typically* expected to have a lower growth efficiency than slow-growing oligotrophs. This is thought to be because of several reasons: 1. there's a general thermodynamic trade-off between ATP yield and ATP synthesis rate (Pfeiffer et al. 2001, 10.1126/science.1058079) which translates into CUE vs growth rate trade-off. 2. copiotrophs typically (though not always) have larger genomes (= higher carrying costs). 3. Oligotrophs often lack energy-costly functions such as motility and protection from ROS. Oligotrophs also typically have slower turnover of macromolecules. 4. Slower growth allows balancing internal redox states and better matching of enzyme production with available resources (both external and internal intermediates, e.g. Baldazzi et

al. 2023, doi.org/10.7554/eLife.79815, Flamholz et al. 2025, 10.1073/pnas.2404048121). A seminal review (and hypotheses) on the subject, which is cited in the revised version of our MS, was written by Roller and Schmidt (2015).

**Reviewer 2**

The manuscript describes the study of microbial carbon use efficiency under different aeration conditions and mineral characteristics using two soils from Bohemian Forest (Czechia) sites. The topic and presented results are interesting and valuable. However, the presentation quality should be improved substantially before the publication. Thus, overall revision and sentence refining are essential throughout the manuscript. The following are specific comments:

Thank you for your comments. We have revised the manuscript and refined some statements.

L69-L70: What did you mean?

Lines 69 and 70 have been removed.

L93-L95: Brief introductions of RNA-SIP with proper references are essential.

RNA-SIP is a fairly standard method in microbial ecology and is described in the methods. Nevertheless, we have added the following sentences and citations to the introduction. "RNA-SIP utilises the incorporation of a heavy stable isotope (13C) into the rRNA of active microorganisms to separate the labelled RNA from the non-labelled one using buoyant density ultracentrifugation in a density gradient. Following fractionation, reverse-transcription and DNA sequencing the identity of the labelled microorganisms is confirmed by statistically comparing their abundance across the different fractions (Angel, 2019; Ghori et al., 2019)."

L109: Soil classification name in FAO/WRB or USDA Soil Taxonomy is essential.

We added the soil classification to the methods. The soils at both sites are cambisols

L111: Table S2 should presented in the main text not as supplementary.

Table S2 only provides background information and most of the values are mentioned in the text. Therefore, we decided to keep this table in supplementary material.

L111: What is "mM Fetot kg-1"? "mM" should be "mmol/L". Correct to proper unit and values.

Corrected to mmol kg-1 here and elsewhere.

L113-L114: Soils with nearly 50% of carbon concentrations should be organic soils in organic layers (i.e., O horizon) but not in mineral soil layers, such as A horizon. At least, the authors should clarify their soil samples were taken from upper organic horizons or the uppermost organo-mineral horizon (Oh + Ah) as mentioned in Kana et al. (2019). Also, they should mention the thickness of organic layers which covered the soil samples.

Thanks for the comments – we should have been more precise. We revised the text:

"The uppermost forest floor layer (2-5 cm) was removed and organo-mineral  (A, Oh+Ah) soil was sampled to a depth of 5 cm at four randomly chosen locations. Fresh soils were pooled, sieved (2 mm) and homogenised prior to storage at 4 °C for several days before the experiments."

Unfortunately, in the PL site, the O horizon runs very deep and mixes with the A horizon. We removed the litter and then sampled at an identical depth. Luckily, the microbial biomass was similar in both samples so the results are unlikely to be affected.

L115: Did you never remove plant roots and debris, which usually cause unexpected variations within the samples?

We did. The soils were sieved using a 2 mm mesh, which removes all larger roots. The fine roots, however, would remain in soil. We performed the pre-incubation step to ensure that the fine roots mostly decayed before glucose addition.

L152: "CO2 and O2" should be correct.

Corrected

L237: Why the authors never analyzed fungal community? In such organic-rich soils, fungi should contribute substantially on organic matter decomposition. And, is the target of used primer sets 16S ribosomal RNA gene? Please clarify.

Fungi certainly contribute to C cycling in these soils, although their role in iron reduction is still unclear. Importantly, an analysis of the fungal community would mean doubling our sequencing efforts and costs and shifting our method to the less sensitive DNA-SIP since this would require sequencing the ITS gene (which is not transcribed). In addition, soil handling and homogenization destroyed fungal hyphae affecting their growth during the experiment. The effect of handling on prokaryotes should not be so strong.

L315: What was the detection limit of GC for oxygen?

1000 ppmv (now added to the methods).

L319-L321: Clarify whether this sentence for oxic or anoxic.

Anoxic conditions. This was corrected.

L323-L325: Did you measure 13C in total SOC after the complete of the incubation? It should be essential to confirm non-occurrence of unconsidered leak of C from the incubator.

Yes we did, but the addition of $^{13}C$ in glucose ($2.5 – 2.9$ µmol $^{13}C$ $g^{-1}$ dry soil) was three orders of magnitude lower than that of total soil $^{13}C$ (411 and/or 155 µmol/g dry soil for PL and CT soils). Thus SOC could not be used for control of leakage. We added the information about $^{13}C$ in SOC into the supplemental (Table S2) and L. 325-327.

L334: Is Table S4 correct? (I think referring to "Table S3" is wrong) Furthermore, Table S4 should be presented in the main text not as supplementary.

Yes, we refer to Table S4. Corrected. The table has been moved to the main text.

L339: "e-acceptors" should be revised to "electron acceptors".

Corrected.

L347-L351: Sentences are likely duplicated. Revise those sentences carefully. Also, "it was vice versa" is very confusing. Describe more specifically.

Corrected.

L352 (Figures 1 & 2 and their captions): Keep the figure and its caption properly close.

Corrected.

L368-L408: Redundant and unclear sentences and construction make the paragraph difficult to understand. Revise those sentences thoroughly.

Corrected.

L372: Fig. S2 should be presented in the main text, not as supplementary. Multiple references to the same supplementary material should be avoided.

According to the suggestion of the other reviewer, we simplified this part of the results and we refer to the CUEs calculus in the early stage of incubation (the results were biased with large error in the early incubation period) only in the supplementary material. Anyway, due to the large error, the results could not be interpreted reliably.

L372-L374: Unneeded sentence. Remove this.

Corrected.

L373-L374: Is Table S3 correct? (I think referring to "Table S1" is wrong)

The reference was correct, but in any case, the sentence was removed.

L374-L376: Should be in a single sentence.

The sentences have been revised for clarity.

L378: Should be Table 2 and Fig S2? (I think referring to "Fig. 1" is wrong)

Corrected.

L415-L422: These sentences should be moved to the materials and methods.

We disagree. These sentences describe the sequencing results (albeit trivial) not the methods.

L428: Revise to "Fig. 2, also see Table S8 and Figs. S5-S8".

Corrected.

L468-L597: Through the discussion, the authors should pay more attention to Indicate figure and/or table numbers which shows the results regarding the sentence here.

References to the figures and tables were added to the discussion.

L469-L473: It is very unclear why the authors placed the sentences here. This should be moved to the introduction or the materials and methods.

This information is of course also mentioned in the introduction. However, we feel that a brief recap is needed here before the sentence starting with "The amount of added glucose was intentionally small..."

L517: Why did the authors conduct nothing to measure 13C in total SOC after the completion of the incubation?

We measured $^{13}$C-SOC also at the end of incubation, but, as mentioned above, the $^{13}$C addition was too small and change in total $^{13}$C-SOC was within analytical error.

In addition to those above issues, self-revising and refining the sentences throughout the manuscript are strongly recommended.

The language has been revised

**Reviewer 3**

General comments:

I read the text with interest. Maybe I am not that deep in the subject, but it seems to me a helpful study for a better understanding of CUE. In some formulations the authors could stay closer to the research, e.g. the title, but overall I am in favour of the publication of this work.

Thank you for your input. Please see our replies below.

specific comments:

Perhaps a title that makes it clear that CUE is not solely determined by the two factors analysed would be good.

That's a good point. We revised the title to: Aeration and mineral composition of soil mediate microbial CUE.

Table 1: please check the unit for Ctot, the content is very high so it would be litter rather than Ah horizons. And a clearer assignment of units to columns is needed.

Indeed, at the PL cite the O horizon is very deep and mixes with the A horizon. We removed the litter and then sampled at an identical depth. Luckily, the microbial biomass was similar in both samples so the results are unlikely to be affected.

Fig 1 A: Is the SEM for glucose at time 0 too low to be displayed? A note to this effect would be good, as it looks as if it has not been measured but only estimated, which is not correct.

The glucose in time 0 was not measured in the soil. It reflects the amount that was added. This is now clarified in the legend.

lines 349-351 are doubled.

Corrected.

L362 microbial glucose uptake stops but not microbial growth

True. This has been corrected.

lines 392-408 are doubled.

Corrected.

L457 and 458: otherwise the site name is always abbreviated to PL

Corrected.

L477 I do not agree with the last part of the statement in this generalised form.

The phrase: " and thus is unlikely to have caused a significant disturbance" has been replaced with "mimicking natural fluctuations in available C"

L564 As the two soils certainly differ in other characteristics, I am not convinced that this is due to the iron content, perhaps more to considerable or substantial.

The sentence has been removed.

---

## Author Response (AR2)

**Editor**

Additional private note (visible to authors and reviewers only):

The background in the revised abstract is way too long. It covers half the text. This text is repeated in the introduction section. Please articulate the rationale for the study more succinctly in the abstract. The abstract should be equally sectioned into background/aims, approach/methods, results, and discussion/conclusions/implications.

Dear Ashish Malik,

Thanks again. The abstract has been revised. The background has been shortened and the results extended.